# Damage Monitoring of Braided Composites Using CNT Yarn Sensor Based on Artificial Fish Swarm Algorithm

**DOI:** 10.3390/s23167067

**Published:** 2023-08-10

**Authors:** Hongxia Wang, Yungang Jia, Minrui Jia, Xiaoyuan Pei, Zhenkai Wan

**Affiliations:** 1Engineering Teaching Practice Training Center, Tiangong University, Tianjin 300387, Chinajiaminrui@tiangong.edu.cn (M.J.); 2National Experimental Teaching Demonstration Center for Engineering Training, Tianjin 300387, China; 3Tianjin Branch of National Computer Network Emergency Response Technical Team/Coordination Center of China, Tianjin 300100, China; 4School of Textile Science and Engineering, Tiangong University, Tianjin 300387, China; peixiaoyuan@tiangong.edu.cn

**Keywords:** carbon nanotube yarn (CNT yarn), artificial fish swarm algorithm (AFSA), optimized configuration of sensors, braided composites, damage location source

## Abstract

This study aims to enable intelligent structural health monitoring of internal damage in aerospace structural components, providing a crucial means of assuring safety and reliability in the aerospace field. To address the limitations and assumptions of traditional monitoring methods, carbon nanotube (CNT) yarn sensors are used as key elements. These sensors are woven with carbon fiber yarns using a three-dimensional six-way braiding process and cured with resin composites. To optimize the sensor configuration, an artificial fish swarm algorithm (AFSA) is introduced, simulating the foraging behavior of fish to determine the best position and number of CNT yarn sensors. Experimental simulations are conducted on 3D braided composites of varying sizes, including penetration hole damage, line damage, and folded wire-mounted damage, to analyze the changes in the resistance data of carbon nanosensors within the damaged material. The results demonstrate that the optimized configuration of CNT yarn sensors based on AFSA is suitable for damage monitoring in 3D woven composites. The experimental positioning errors range from 0.224 to 0.510 mm, with all error values being less than 1 mm, thus achieving minimum sensor coverage for a maximum area. This result not only effectively reduces the cost of the monitoring system, but also improves the accuracy and reliability of the monitoring process.

## 1. Introduction

As an advanced composite material, three-dimensional (3D)-braided composites have become important structural materials used in the aviation and aerospace fields due to their high specific strength, high specific modulus, high damage tolerance, and other excellent properties [1,2]. Moreover, 3D composites have complex fiber structures. However, many factors, such as the braiding process, structural parameters, mechanical properties of braided yarn and matrix, void ratios, and interface damage, have important impacts on their structure and mechanical properties [3,4,5]. Indeed, 3D braided composites’ parts will suffer various stresses and strains in the process of application, resulting in structural damage, which may lead to various serious accidents and cause huge losses of life and objects and environmental pollution [6,7,8,9]. As the shape of a 3D braided composite structure hardly changes before failure, it becomes difficult to detect any internal damage through common visual or knock inspections [10,11]. Under the premise of long-term operation of composite materials, in order to ensure the safe use of the structure, it is very important to implement full-life and online intelligent structural health monitoring from the beginning of the manufacturing period to the end of the structure’s life [12].

Structural health monitoring (SHM) refers to the use of on-site non-destructive sensing technology to detect structural damage or degradation by analyzing structural system characteristics, including structural response [13,14]. The purpose of SHM is to significantly improve the safety of materials by constantly monitoring the damage caused to the material structure during its whole service life [15,16,17]. At present, the SHM methods for 3D braided composites include visual inspection, X-ray inspection, acoustic emission inspection, ultrasonic inspection, eddy current, linear scanning inspection of pure S0 and SH0 waves, and fiber Bragg grating inspection [18,19,20]. However, these methods have great inconveniences and limitations. For example, these detection methods usually require the use of complex equipment, online real-time monitoring is difficult, and the damage and location of the tested structure need to be preliminarily understood. The Bragg grating sensor has the advantages of low cost, no electromagnetic interference, and be able to monitor the internal changes in the structure [21]. Therefore, some researchers consider embedding the Bragg grating sensor into the composite to detect the damage caused to the composite. However, the optical fiber of the optical fiber sensor is brittle, meaning that it has poor braiding ability and is not easy to connect [22,23,24]; thus, it is difficult to conduct the braiding process. CNT yarn sensors have unique advantages, such as small size, excellent mechanical properties, high sensitivity, and easily being able to build sensor networks [25]. It is found that adding CNT into the preform of 3D braided composites can improve the mechanical properties of composites to a certain extent [26]. In addition, the strain characteristics of carbon nanowire sensors can be used to monitor the stress and strain of 3D braided composites in real time, thus realizing the purpose of structural health monitoring of 3D braided composites [27].

Son, W. et al. [28] developed a yarn-type hydrogen gas-sensing platform (HGSP) by combining palladium oxide nanoparticles (PdO NPs) and spinnable CNT buckypapers. The results show that the biscrolled HGSP yarn exhibits a rapid response time and high sensitivity to flammable H_2_ concentrations, and it can even detect H_2_ gas leakage in curved and jointed areas in real time. Zhang et al. [29] conducted a review of carbon-based flexible wearable sensors and their fabrication, performance, and working mechanisms. The study concluded that CNTs and graphene can be added to various macrostructures to create flexible sensors in the form of films, fibers, yarns, or fabrics.

With the increase in the material size and the improvement in the requirements for the accuracy of structural health monitoring, the number of CNT yarn sensors is increasing. As a result, the number and weight of cables and supporting equipment connected to the CNT yarn sensor are also increasing. The following question is, thus, considered: how can we embed the CNT yarn sensor into a reasonable position in the material, reduce the number of sensors as much as possible, and reduce the overall weight of the material? Another important question is as follows: how can we improve the coverage and sensitivity of sensor response information, effectively monitor the information of three-dimensional braided composite structure status, and achieve the purpose of safety assessment of material health status? These are important topics to consider in structural health monitoring [30]. 

For the optimal configuration of sensors, based on different algorithms, many schemes have been proposed by researchers. The traditional optimal allocation algorithm has the disadvantages of requiring sufficient prior knowledge, making it difficult to deal with the target noise, and poor robustness, which lead to large errors in the optimal allocation scheme and cannot meet the accuracy required for optimal allocation. With the continuous emergence of intelligent optimization algorithms, ant colony optimization (ACO), particle swarm optimization (PSO), the genetic algorithm (GA), the neural network algorithm, the non-dominated neighborhood immune algorithm (NNIA), and the artificial fish swarm algorithm (AFSA) have gradually become new choices in the field of sensor optimal configuration. Among these choices, the AFSA is a bionic algorithm, which has the advantages of simple implementation, insensitivity to initial values, strong robustness, good global convergence and strong solution performance, and being usable to deal with the complex, high-latitude, and multivariable optimization problem of sensor position optimization [31]. Compared to GA, AFSA avoids the complex encoding and decoding process, and the population generated via each iteration is more directional. Compared to PSO, the AFSA prevents premature phenomena to a certain extent and has looser requirements for parameters. It has the characteristics of fast iterative search for the optimal solution [32]. Therefore, in practical engineering, the AFSA is widely used.

Daneshvar et al. [33] proposed a method that aimed to locate and quantify damage by developing a new sensitivity function based on modal strain energy and solving an ill-posed inverse problem using an optimization-based iterative regularization method called iteratively reweighted norm-basis pursuit denoising (IRN-BPD). The results demonstrate the success of the methods in locating and quantifying damage, even with incomplete and noisy modal data. Miao et al. [34] used AFSA to optimize the multi-information fusion fire prediction model based on a back-propagation neural network, which accelerated the convergence speed of the model, improved the prediction accuracy, and obtained the optimal fire prevention decision. Sun et al. [35] proposed an internal double-layer-shielded soft magnetic strip structure optimization method based on the artificial fish swarm algorithm and applied it to the coupler of the electric vehicle wireless-charging system. A group of parameters with the largest coupling coefficient and appropriate volume are obtained through the AFSA. The optimal transfer efficiency can be achieved without employing a lot of repeated finite element analysis results. In order to improve the operation efficiency levels and service lives of wireless sensor networks, Feng et al. [36] adopted the improved artificial fish swarm algorithm to optimize the coverage of wireless sensor networks, which not only ensured the quality of network service, but also achieved maximum network coverage and minimum energy consumption. Huang et al. [37] proposed an optimized layout method of a fiber Bragg grating sensor network based on an improved artificial fish swarm algorithm, which optimized the sensor layout and improved the convergence accuracy, speed, and network coverage of the algorithm. Kou et al. et al. [38] proposed a combined fault diagnosis method based on an improved and fully integrated EMD energy entropy and support vector machine and optimized it using AFSA. The diagnosis accuracy of the optimized fault diagnosis model improved to 97.5%. Chen et al. [39] developed a dynamic liquid level error compensation model for a soft sensor using a Gabor kernel limit learning machine optimized using an artificial fish swarm algorithm. Their aim was to enhance the accuracy of the mechanism model. In order to optimize the 3D target coverage of a underwater heterogeneous multi-sensor, Zhou et al. [40] proposed a chaotic parallel artificial fish swarm algorithm (CPAFSA) to improve the coverage of underwater sensors and optimize the water quality-monitoring sensor network. CPAFSA uses chaos to select initialization parameters, adopts elite selection, effectively avoids local optimization, and integrates the global search function of parallel operators to solve the three-dimensional target coverage problem. Meng et al. [41] applied the AFSA to robot path planning in an intelligent bionic manner. This application improved the solution speed of the planning, reduced the length of the planned path, decreased the convergence speed of the algorithm in the later stage, and facilitated the attainment of the local optimal solution. It can be observed from the above-mentioned study that the artificial fish swarm algorithm plays a significant role in optimizing sensor coverage allocation.

The fibers in the 3D braided composites are interwoven according to certain rules, thus forming a “spatial network” structure with interlaced fiber bundles in multiple directions, which makes it particularly difficult for the sensor to monitor the status of any internal damage. However, a reasonable sensor arrangement is the prerequisite for the health monitoring of braided composite structures. The optimal sensor arrangement scheme can reduce the required number of sensors, improve the coverage and sensitivity of response information, and reduce the weight of braided composites. In this work, with the aim of resolving the problem of online monitoring and damage location of braided composite structures, the CNT yarn sensor is embedded in braided composites in the form of axial yarn and six-way yarn to build a lightweight and high-strength sensor with a controllable network structure. The mechanical and electrical response characteristics of the CNT yarn sensor are utilized to realize sensitive online monitoring of the structural health of braided composites. The AFSA is used to analyze the optimal configuration of the number and position of CNT yarn sensors. This approach provides a theoretical basis for the design and development of intelligent 3D braided composites that can monitor the structure’s health online.

## 2. Materials and Methods

### 2.1. Analysis of CNT Yarn Sensors and Their Characteristics

CNTs are a novel type of high-performance fiber material. CNT wires, which are composed of CNTs, are formed through twisting, bonding, polymer wrapping, and energetic particle beam welding, among other measures, resulting in high strength. Due to their exceptional toughness, the mechanical properties of CNT wires are not affected by repeated bending, knotting, or uneven stress distribution. As a result, they are effectively utilized in areas such as bending areas, contact points, and areas with uneven stress, effectively compensating for the low interfacial strength and high brittleness of ordinary carbon fiber-reinforced composites. The CNT thread, which was made via dry spinning, exhibited such excellent mechanical and electrical properties through the processes of twisting and winding [42].

The 3D woven composite pre-fabricated parts are made of woven carbon fiber yarns, which have a certain level of electrical conductivity. The embedded CNT yarns have even better electrical conductivity. When the two materials are in close contact, this contact leads to a significant change in the measured resistance of the CNT yarns. Additionally, the diameter of the CNT yarn is only between tens and one hundred micrometers, making it susceptible to loss during the embedding process. Therefore, the CNT yarns need to be processed before they are embedded as sensors in 3D woven composites.

Firstly, an insulating varnish was applied to the outer layer of the embedded part of the CNT yarn to create an insulating protective layer. Then, conductive silver adhesive was applied to the exposed ends of the CNT yarn to form end connectors. These end connectors ensured a reliable connection between the CNT yarn sensor and the measuring device, as shown in Figure 1.

The CNT yarn used in the processing treatment was developed by Suzhou Hengqiu Graphene Technology Co., Ltd., which is situated in Suzhou City, Jiangsu Province, China. It was prepared via dry spinning of CNT arrays, which are multi-walled CNT fibers. Table 1 provides the performance control figures of different models of carbon nanowires that were successfully developed by the company. 

The resistance of CNTs monotonically increased with increasing pressure. When subjected to tensile force, the resistance of carbon nanowires was influenced by two factors: the contact radial force generated by the nanotubes, which increased the contact between adjacent CNTs, and the pressure applied to individual nanotubes within the carbon nanowire [43]. Since increasing contact would decrease resistance, the first factor could not account for the actual increase in resistance. Therefore, the increase in resistance occurred due to the compression of individual CNTs within the carbon nanowire as pressure increased [44].

According to Ohm’s law,
(1)$$R=\frac{\rho L}{A}$$

In the equation, *R* represents impedance, *L* represents the length of CNTs, ρ represents resistivity, and *A* represents the cross-sectional area of CNTs.

The change in resistance of CNTs was calculated using Equation (2):(2)$$\frac{∆R}{R}=\frac{∆\rho }{\rho }+\frac{∆L}{L}-\frac{∆A}{A}$$

The change in the length of CNTs was described using Equation (3):(3)$$\frac{∆L}{L}=\varepsilon_{11}-\frac{\varepsilon_{11}^{2}}{2}$$

$\varepsilon_{11}$ denotes the engineering strain, which is defined as the unit length change in the material. If the deformation of CNTs is small, the second term in the equation can be neglected. According to the theory of Poisson’s effect, the cross-sectional change in CNTs is transversely isotropic, so the sensor cross-section could be represented using Equation (4):(4)$$A^{\prime }=A\left(1-2v_{12}\varepsilon_{11}\right)$$
which is the Poisson’s ratio of CNTs. Therefore,
(5)$$\frac{∆A}{A}=-2v_{12}\varepsilon_{11}$$

Therefore,
(6)$$\frac{∆R}{R}=\frac{∆\rho }{\rho }+\varepsilon_{11}+2v_{12}\varepsilon_{11}=\frac{∆\rho }{\rho }+\varepsilon_{11}\left(1+2v_{12}\right)$$

Due to the symmetry of material load-bearing and relative resistivity increments, carbon nanowire sensors exhibited good linearity within a certain load range. The above analysis of carbon nanowire sensor characteristics provided a theoretical basis for constructing a three-dimensional woven composite material damage detection system based on carbon nanowires.

### 2.2. Insertion of CNT Yarn Sensor

Three-dimensional woven composites embedded in CNT yarn sensors consisted of three parts: a CNT yarn sensor, a carbon fiber yarn, and a resin. The carbon fiber yarn was formed by combining a certain number of fiber bundles, and there were certain voids between the fiber bundles. During the curing process, the resin was fully infiltrated into the voids of the fiber bundles. Moreover, 3D braiding technology is a kind of multi-directional near-net forming pre-form braiding technology. In braided composites, the braided yarn is bent. In order to realize the accurate monitoring of the internal damage of braided composites, the problem of monitoring error caused by yarn bending had to be solved. Indeed, 3D six-direction braided composites are a new type of composite material that has strong structural integrity and excellent mechanical properties. This composite is produced using the 3D four-direction braiding process, with the addition of an axial fixed yarn (referred to as axial yarn) in the fifth direction and a weft fixed yarn (referred to as weft yarn) in the sixth direction, which is perpendicular to the braiding direction. In the 3D six-direction braided composites, the spatial orientation of the yarn was structured as shown in Figure 2. Weft yarn will not bend during weaving, so in order to reduce the signal data error caused by CNT yarn bending, CNT yarn was embedded in the pre-form of braid composites as weft yarn during braiding, so that CNT yarn was “linear” inside of the braided composites [45]. Global monitoring of internal damage in braided composites was achieved by designing the number and location of embedded CNT yarns.

Yarn carriers were evenly distributed on the chassis of the 3D braiding machine, with ○ representing the braid yarn carrier and □ representing the axial yarn carrier, as shown in Figure 3. During the braiding process, first of all, we fixed one end of the braiding yarn and the shaft yarn to the holder above the chassis. The other end of the yarn was fixed to the braid yarn carrier and the axial yarn carrier of the chassis based on the cross-sectional shape of the 3D braided composites. The braid yarn was carried by the yarn carrier and moved in a certain manner in the direction of rows and columns. The yarns were interwoven above the braiding machine, realizing the entire braiding process. Braiding yarn underwent four movement steps and then returned to its initial position, which represented a mechanical cycle. Therefore, it was also known as the “four step method” of braiding. The specific braiding process used in this study is shown in Figure 3.

During the braiding process, the axial yarn carrier only reciprocated in the direction of the row, while the weft yarn shuttled back and forth in a direction perpendicular to the axial yarn, spanning the overall width of the braid fabric. Neither the axial yarn nor the weft yarn was involved in the braiding process. Instead, they were evenly sandwiched in the gap between the woven yarns in the braiding–molding direction and the width direction, respectively. Therefore, sensors could be added in the axial and weft directions to monitor the internal damage of woven composites. The CNT yarn sensor embedded in the axial yarn direction was called an axial sensor. To complete a machine cycle, it was necessary to perform an alignment “tightening” operation on the braiding yarn and the spindle yarn. At this time, a CNT yarn sensor was inserted along the weft direction of the preform to replace the weft yarn, achieving the insertion of the sensor in the weft direction. The CNT yarn sensor embedded in this direction is called a weft sensor. The embedded position of the sensor is shown in Figure 4 and Figure 5. 

### 2.3. CNT Yarn Embedding Model Based on AFSA

AFSA is a swarm intelligent optimization algorithm that simulates the characteristics of fish schools. The basic idea is to utilize the characteristics of fish schools swimming towards waters with high food concentrations and find the region with the highest food concentration based on a series of behaviors demonstrated by fish schools searching for food to obtain the global optimal solution to the problem [46]. This method has the advantages of fewer design parameters, easy implementation, and a fast optimization speed. In the process of optimizing the configuration of CNT yarn sensors, the issues that need to be addressed are the number and location of embedded axial and weft sensors. Assuming that the CNT yarn sensor node set is $C=\left\{c_{1},c_{2},c_{3},\ldots ,c_{N}\right\}$, we found a subset $\left|C^{\prime }\right|$, which maximized coverage $R_{cov}\left(C^{\prime }\right)$ and minimized the number of nodes $\left|C^{\prime }\right|$. The flowchart depicting the process of embedding the CNT yarn sensors via a model based on AFSA is shown in Figure 6. 

#### 2.3.1. Node Coverage

The monitoring area was then set as *E*, and CNT yarn sensor nodes with the same parameters were placed in this area with a number of *N*. If the coordinates of each node were known and the effective monitoring radius was *r*, the sensor node set was represented as $c=\left\{c_{1},c_{2},c_{3},\ldots ,c_{N}\right\}$, where, $c_{i}=\left\{x_{i},y_{i},r\right\}$ represented a circle with a monitoring radius of r and a node coordinate $\left\{x_{i},y_{i}\right\}$ as the center of the circle. Assuming that the monitoring area *E* was digitally discretized into m×n pixels, where the event at which any pixel point $S\left(x,y\right)$ was covered by a CNT yarn sensor node i was defined as $r_{i}$, the probability $P\left\{r_{i}\right\}$ of this event occurring was the probability $P_{\mathrm{cov}}(x,y,c_{i})$ that the pixel point $\left(x,y\right)$ was covered by the sensor node i [47], as shown in Figure 7.
(7)$$P\left\{r_{i}\right\}=P_{cov}\left(x,y,c_{i}\right)=\left\{\begin{array}{l} 1,f((x-x_{i}{)}^{2}+(y-y_{i}{)}^{2})\leq r^{2}\\ 0,or\;else \end{array}\right.$$

Equation (7) indicates that when the distance between pixel point $S\left(x,y\right)$ and CNT yarn sensor node i is less than the sensing range r, the pixel point $S\left(x,y\right)$ is considered to be covered by CNT yarn sensor node i.
(8)$$P\left\{\bar{r_{i}}\right\}=1-P\left\{r_{i}\right\}=1-P_{cov}\left(x,y,c_{i}\right)$$

In Equation (8), $\bar{r_{i}}$ is the complement of $r_{i}$, indicating that the pixel point $S\left(x,y\right)$ is not covered by the CNT yarn sensor node i. If $r_{i}$ and $r_{j}$ are unrelated, the following relationship exists:(9)$$P\left\{r_{i}\cup r_{j}\right\}=1-P\left\{\bar{r_{i}}\cap \bar{r_{j}}\right\}$$

Node set means that as long as one node covers pixel point $S\left(x,y\right)$, the pixel point $S\left(x,y\right)$ is considered to be covered by the CNT yarn sensor node set. Therefore, the probability that a pixel point $S\left(x,y\right)$ is covered by a node set is the union of $r_{i}$. Assuming that all random events $r_{i}$ are independent of each other, the coverage of node set *C* can be calculated using Equation (10):(10)$$\begin{array}{c} \;P_{cov}\left(x,y,c\right)=P\left\{\begin{array}{c} N\\ Y\\ i=1 \end{array}r_{i}\right\}=1-P\left\{\begin{array}{c} N\\ I\\ i=1 \end{array}\bar{r_{i}}\right\}=1-\prod_{i=1}^{N}\left[1-P_{cov}\left(x,y,c_{i}\right)\right] \end{array}$$

In Formula (10), if all CNT yarn sensor nodes do not cover pixel point $S\left(x,y\right)$, pixel point $S\left(x,y\right)$ is considered to be an uncovered point. Otherwise, the pixel point $S\left(x,y\right)$ is considered to be covered by the CNT yarn sensor node set.

#### 2.3.2. Area Coverage

There were m×n pixels in the monitoring area *E*, and the area size of each pixel could be expressed as △x × △y (assuming that the area of each pixel was 1). Whether each pixel was covered was measured using the coverage rate $P_{\mathrm{cov}}(x,y,c_{i})$ of the CNT yarn sensor node set, and the area coverage rate $R_{area}(C)$ of the CNT yarn sensor node set *C* was defined as the ratio of the coverage area $E_{area}(C)$ of the CNT yarn sensor node set *C* to the total area $E_{S}$ of the monitoring area *E*, as shown in Equation (11):(11)$$R_{area}(C)=\frac{E_{area}(C)}{E_{S}}=\frac{\sum_{x}^{m}\sum_{y}^{n}P_{\mathrm{cov}}(x,y,C)}{m\times n}$$

### 2.4. Process of AFSA

We supposed that each artificial fish represented a CNT yarn sensor, and the coordinates of the artificial fish were the locations of the CNT yarn sensor.

(1) Initialize the school of fish: 

The initial state assigned to a school of fish using integer coding was $X=\left(x_{1},x_{2},\ldots ,x_{n}\right)$, the process state of the school of fish was $X_{k}=\left(x_{1k},x_{2k},\ldots ,x_{nk}\right)$, the initial coordinate state of the artificial fish was $M_{0}=\left\{\left(x_{10},y_{10}\right),\left(x_{20},y_{20}\right),\ldots \left(x_{n0},y_{n0}\right)\right\}$, and the objective function was $Y=f\left(x\right)$.

(2) Foraging behavior:

We set the current artificial fish position as $M_{0}\left(x_{0},y_{0}\right)$, and the distance to its nearest artificial fish was set as $d_{min}$. Artificial fish randomly determined a point N within the maximum field of view.

If the minimum distance from point N to artificial fish other than point $M_{0}$ was greater than $d_{min}$, the food concentration at point N was considered to be superior to the food concentration at point $M_{0}$. The current artificial fish moved in steps in the direction of the N point. If not all of the conditions were satisfied, the system randomly moved one step, with Rand () being a random number between 0 and 1 and the degree of congestion being δ.

(3) Reaching behavior:

We supposed the distance from the current artificial fish $M_{0}F$ to the nearest artificial fish NF was $d_{min}$. If $d_{min}>2\times distance$, we moved forward one step in the direction of NF, and if $d_{min}<\left(\frac{\sqrt{3}}{2}\right)\times distance$, we moved forward one step in the opposite direction of NF. If none of the conditions we satisfied, we randomly move one step.

(4) Random behavior: 

Artificial fish randomly moved one step within the field of vision in a random step size, which was the default behavior for tail chasing and foraging.

(5) Group bulletin board: 

After each artificial fish successively performed tail-chasing and foraging behaviors, its location information was written on a bulletin board and stored.

(6) Coverage determination:

We calculated the distance from each pixel point to the artificial fish. Once it was less than the detection distance, it was considered to have been covered. The current coverage was calculated using Equation (11). If the coverage requirement (threshold) was met, the iteration was terminated. If the coverage requirements were not met, we continued the next iteration until the coverage requirements were met.

### 2.5. Simulation Experiment of Optimal Configuration of CNT Yarn Sensor

Three-dimensional woven composite materials are flat structures. In this study, a 12 × 10 grid node was used as an example, being represented by dots, to indicate the potential occurrence of damage signals at all nodes. Figure 8 illustrates this fact. Based on the optimization criteria of sensor configuration using stress signal monitoring, the AFSA was employed to optimize the placement of the minimum number of sensors in the most suitable positions. This process ensured the maximum monitoring rate of the sensor network for 120 stress signal nodes, resulting in the highest coverage.

We selected 35 carbon nanowire sensors as artificial fish and used AFSA to optimize their configuration. Visual field = 2, step length = 1, try number = 1, and threshold T-value = 0.99, Under the Matlab 7.0 environment, it ran continuously for 30 iterations. When the coverage reached 99.6%, the average number of iterations required was 46. The optimization curve is shown in Figure 9.

As can be seen from Figure 10, the curve shows a downward trend as the number of sensors increases. Moreover, at the beginning of the increase in the number of sensors, there is a sharp decline, which represents a clear trend. As the number of sensors increases to a certain level, the change area becomes gentle. However, when the number of sensors reaches the 48th iteration, the probability of joint monitoring between sensors and network nodes reaches a maximum of 0.99654. However, selecting a relatively large number of sensors to configure is not reasonable from an economic perspective. In order to consider the optimal fitness and economic cost, it is necessary to select an appropriate number of sensors. From Figure 10, it can be found that when the number of sensors increases from the initial configuration of 2 to the 26th iteration, the optimal configuration effect has been achieved, and the change in the sensor curve tends to be flat as the number of sensors increases. Therefore, in ideal conditions, the optimal configuration number of three-dimensional braided composite monitoring sensors for the above 12 × 10 grid node plate structure is 26.

## 3. Results and Discussion

### 3.1. Construction of Damage Detection Systemr

The optimal configuration of CNT yarn sensors can be achieved using the AFSA, though there may be some errors between the calculation results of the algorithm and the actual results of 3D braided composites used in intelligent structural health monitoring. In order to verify the accuracy of the damage location of the embedded CNT yarn sensor, the 3D braided composites specimens embedded in the CNT yarn sensor were prepared. The CNT with a diameter of 25 μm is woven together with T300B-3K carbon fiber produced by Toray Industries Ltd., Tokyo, Japan. Considering that both carbon fiber and CNT have conductivity, to prevent mutual interference, the surface of the CNT should be coated with an insulating layer before weaving. The matrix material used is TDE-86 epoxy resin. The specimen is embedded with CNT yarn sensors using a three-dimensional six-way weaving process and then cured through the vacuum-assisted resin transfer molding (VARTM) process. After the fabrication of the specimen, silver paste is applied to both ends of the CNT. The two CNT endpoints of each sample are connected via wires, and the change in the resistance of the CNT is obtained through a signal acquisition circuit, as shown in Figure 1.

The structure of the 3D braided composites’ damage detection system based on the CNT yarn sensor is shown in Figure 11. The existence and location of damage in 3D braided composites are uniformly monitored, and real-time data are uniformly processed to obtain real-time information, such as the damage status of three-dimensional braided composites, and verify the effectiveness of the configuration scheme derived from the AFSA algorithm.

### 3.2. Experimental Verification of Damage Detection

The damage sensitivity of CNT yarn sensors at fixed locations is different for different locations. The further away the damage is from the sensor, the smaller the impact of load on the sensor, so a reasonable arrangement of carbon nanowire sensors plays a decisive role in determining the accuracy of damage detection. The spacing between CNT yarn sensors should be as small as possible, which can reduce signal loss and facilitate information extraction. However, in practical applications, considering the cost and braiding process of composites, it is not possible to embed a large number of CNT yarn sensors inside of the composites. Therefore, it is necessary to reasonably control the spacing and embedding method of CNT yarn sensors, so as to save resources and effectively detect damage. The number and location of sensors embedded in the woven composite using the AFSA algorithm are shown in Table 2.

Three 3D braided composite specimens embedded with CNT yarn sensors were selected, having dimensions of 180 mm × 180 mm × 10 mm. Each specimen was subjected to damage at different locations, which were referred to as test pieces 1–3, as demonstrated in Figure 12. Test specimens 1 and 2 were each subjected to two damage points. In test specimen 1, there were two through-holes: S1, i.e., a cylindrical through-hole with an upper bottom radius of 0.85 mm and a lower bottom radius of 0.78 mm, and S2, i.e., a cylindrical penetrating circular hole with radii of 0.65 mm on the upper bottom surface and 0.6 mm on the lower bottom surface. In test specimen 2, the damage S3 was a linear damage measuring 4.5 mm in length and inclining at an angle of 15.6° from the horizontal position; damage S4 was a broken line damage with a length of 4.4 mm and inclined at an angle of 43.6° from the horizontal position. Test specimen 3 features a circular damage, which was referred to as S5, with a radius of 0.63 mm.

Due to the relatively large size of the test piece and the large number of embedded CNT yarn sensors, it is difficult to process and analyze the data using manual methods. Therefore, signal denoising and eigenvalue extraction methods are used to process the data. The location of the damage point of the test specimen is clearly described, and the test piece is divided into areas of equal size on the spatial plane. The number of regions to be divided is determined based on the number of embedded CNT yarn sensors. Assuming that the CNT yarn sensors array embedded in the test specimen is m×n, the test specimen is divided into m×n small regions, as shown in Figure 13.

After the braided composites are divided into regions, a feature extraction method combining the whole and the part is adopted. We then block the data matrix according to certain rules and proportions to obtain the eigenvalues and eigenvectors of each sub matrix. Then, the global singular value matrix of the original matrix is combined with the singular value matrix of the local submatrix to obtain a local and global eigenvector.

We also find the following benefits of dividing the data matrix into blocks before processing: For high-dimensional data matrices, directly extracting features from the entire matrix can greatly increase the time complexity of the algorithm and reduce the real-time performance of the system. Moreover, after decomposing the matrix, singular value decomposition is performed on each submatrix. The more submatrices are present, the more information they have, and the better the subsequent recognition effect. To analyze the sensitivity of damage localization, the selected Cartesian plane coordinate system is shown in Figure 13. Assuming that the center point position of the specimen is the origin (0,0), the *x*-axis is parallel to the direction of the row CNT yarn sensors, and the *y*-axis is perpendicular to the direction of the row CNT yarn sensors.

In this experiment, it is necessary to simulate different damage information. By analyzing the changes in the resistance data of CNT yarn sensors inside the material after damage, it is possible to determine the presence of damage and its type and then infer the location of the damage. By grouping the axial and weft CNT yarn sensors and connecting them to a multi-channel selection switch, multiple analog signals can be simultaneously input into a Wheatstone bridge. The resistance change rate signal of the CNT yarn sensor is collected, converted into a digital signal via an analog-to-digital converter (A/D), and uploaded to the computer system after digital processing. The obtained resistance values in the warp and weft directions are shown in Figure 14, Figure 15, Figure 16 and Figure 17. It can be seen from Figure 14, Figure 15, Figure 16 and Figure 17 that the carbon nanowire sensors inside of the healthy specimen have not experienced strain or fracture, and the specimen is not loaded, so the resistivity has not significantly changed. Therefore, the resistance values measured in the warp and weft directions of the healthy specimen are both approximately zero. However, in the other specimens, due to internal damage, the embedded carbon nanowire sensors have experienced strain or even fracture, and the measured resistance data have undergone abnormal mutations.

The wavelet thresholding method and the singular value decomposition method of the quadrature matrix are used to calculate the damage location in specimens 1–3. The actual damage locations are obtained through damage scanning of the specimens. By comparing the eigenvalues to those of the healthy specimens, the presence of damage is determined, and the damage locations are calculated. The maximum coordinate deviation, as shown in Figure 17, is obtained. From the figure, it can be observed that the CNT yarn sensors embedded according to the optimized configuration achieve high accuracy in locating the damage source in larger-sized specimens, with positioning errors all being below 1 mm. This result verifies the correctness of the optimized configuration results.

### 3.3. Comparative Experiment

To verify the performance advantages of the AFSA algorithm in optimization for coverage and monitoring, a comparative experiment was conducted on five test points under the conditions described in Section 3.2. The experiment compared the accuracy errors of damage source localization via CNT yarn sensors embedded using optimization configurations from the AFSA, PSO, and GA algorithms. The experimental results are shown in Figure 18.

From Figure 19, it is clear that the deviation of damage coordinates calculated via the AFSA algorithm is smaller than those of the PSO and GA algorithms. This result indicates that the AFSA algorithm has a strong performance advantage in optimization for coverage and monitoring, as it can improve the positioning accuracy of sensors and help us to better protect and monitor target areas. This result is of significant importance for damage source localization and protection work.

## 4. Conclusions

The AFSA optimization algorithm was employed to address the optimization configuration of CNT yarn sensors. The algorithm successfully solved the optimization problem of sensor quantity and positions for different-sized specimens. Based on the experimental verification, the following conclusions were drawn:

(1) The AFSA algorithm demonstrated excellent optimization capability in the selection of embedding positions for CNT yarn sensors based on the four-step three-dimensional six-directional weaving process. It effectively resolved the optimization configuration problem of CNT yarn sensors and provided optimal configuration results for sensor quantity and positions in different specimens.

(2) In order to accurately describe the locations of damage points in the specimens, unit partitioning and coordinate selection were performed. By collecting the resistance change rate data of CNT yarn sensors, experiments were conducted on intact and damaged specimens to verify the effectiveness of the optimization configuration scheme. These experimental results provided important comparative data for finite element simulations.

(3) In the experiments regarding damage source localization and comparison, the combination of CNT yarn sensors optimized using the AFSA algorithm with the wavelet threshold method and fourth-moment matrix singular value decomposition method accurately localized the damage source, and the experimental positioning error ranged from 0.224 to 0.510, with all values being less than 1 mm. This method exhibited significant effects in optimizing coverage and monitoring, laying a solid foundation for the establishment of an intelligent composite material damage source localization model.

(4) The composite specimens used in the study were produced using a three-dimensional six-directional weaving process, and only flat specimens were subjected to damage detection. Future research could consider the use of a three-dimensional seven-directional weaving process to form a three-dimensional network of sensors inside the structure, enabling the analysis of damage in three-dimensional structures.

In summary, the AFSA algorithm successfully addressed the optimization configuration problem of CNT yarn sensors and achieved accurate damage source localization. These research findings provide important theoretical and experimental foundations for the application and development of intelligent composite materials.

## Figures and Tables

**Figure 1 sensors-23-07067-f001:**
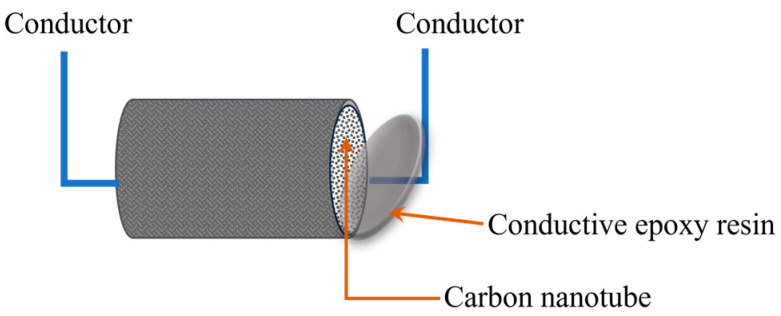
CNT yarn sensor connection diagram.

**Figure 2 sensors-23-07067-f002:**
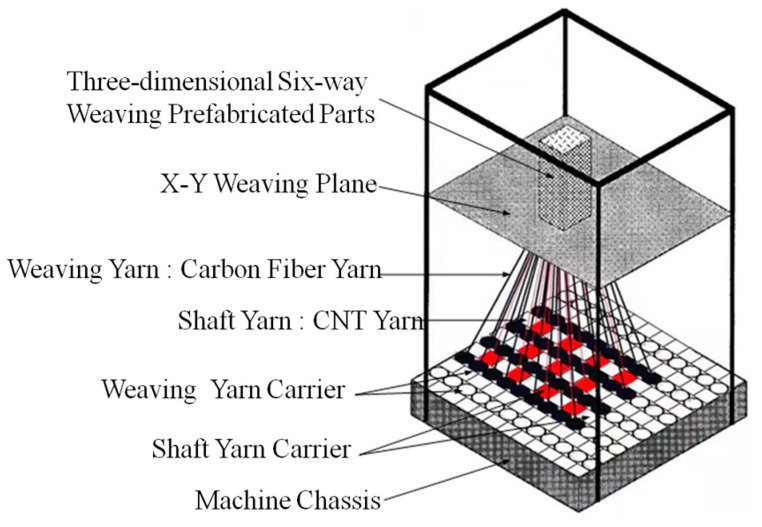
Schematic diagram of three-dimensional braiding machine.

**Figure 3 sensors-23-07067-f003:**
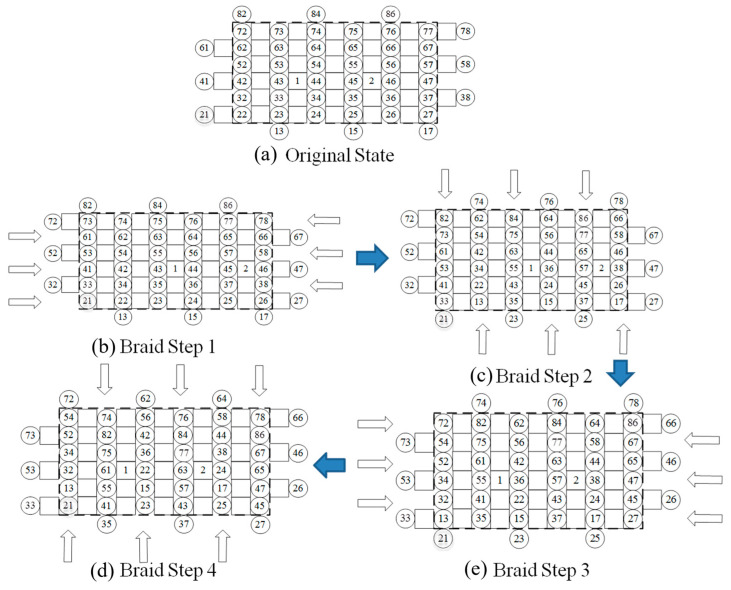
Four-step yarn carriers movement process: (**a**) initial position of yarn carriers; (**b**) Step 1; (**c**) Step 2; (**d**) Step 3; (**e**) Step 4.

**Figure 4 sensors-23-07067-f004:**
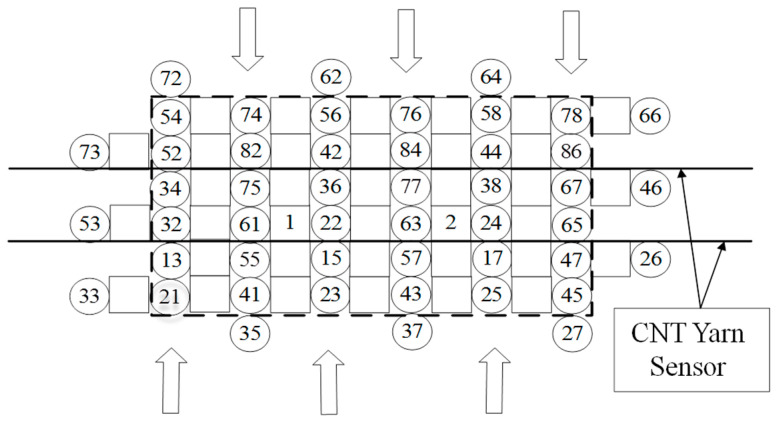
Schematic diagram of filling insertion before fabrication.

**Figure 5 sensors-23-07067-f005:**
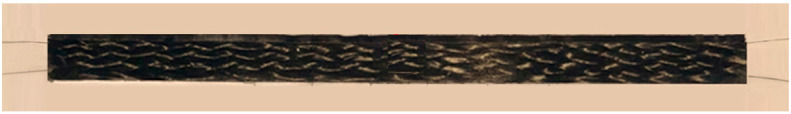
Structure diagram of 3D braided composites embedded in CNT yarns sensors.

**Figure 6 sensors-23-07067-f006:**
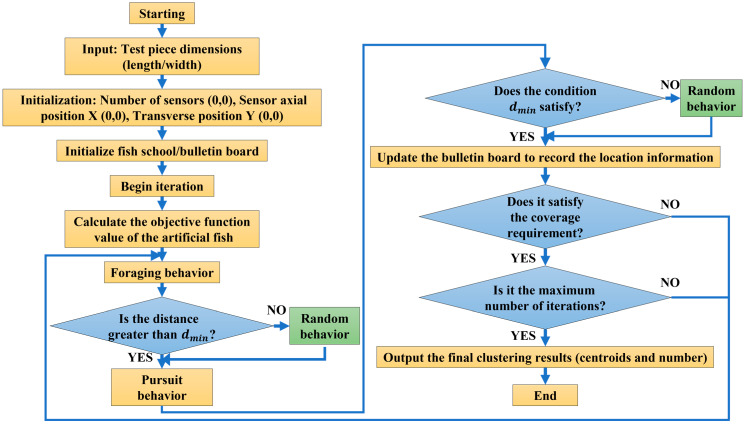
The flowchart of the process of embedding CNT yarn sensor via a model based on AFSA.

**Figure 7 sensors-23-07067-f007:**
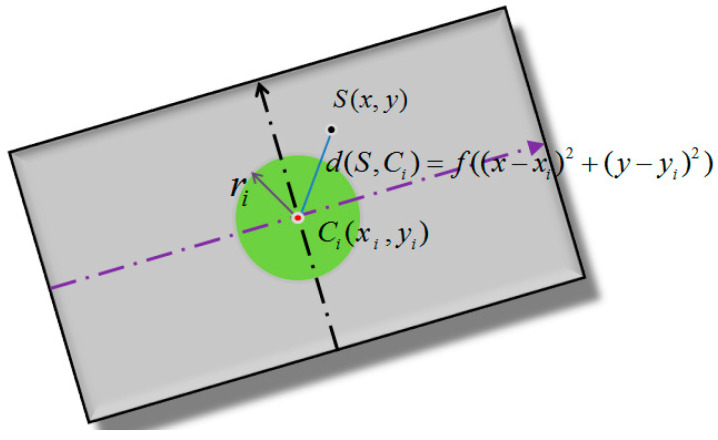
Schematic diagram of binary perceptual model.

**Figure 8 sensors-23-07067-f008:**
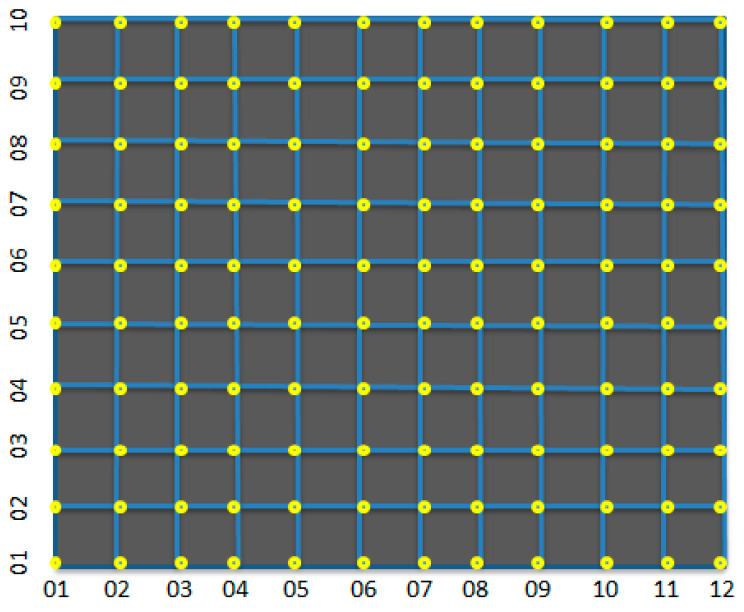
Three-dimensional braided composite with flat structure.

**Figure 9 sensors-23-07067-f009:**
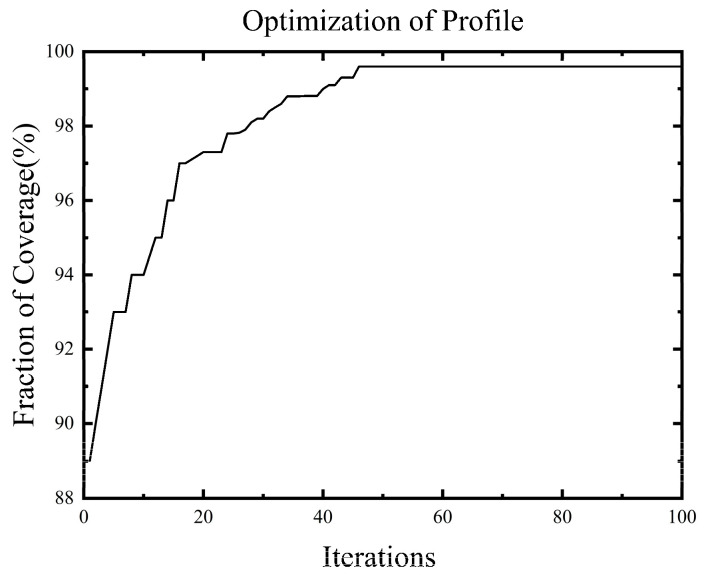
Optimization curve.

**Figure 10 sensors-23-07067-f010:**
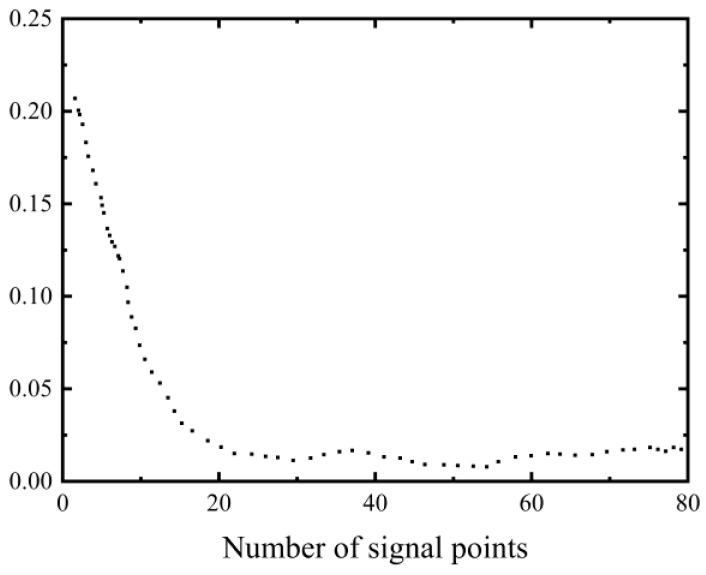
Number of sensors and coverage curve.

**Figure 11 sensors-23-07067-f011:**
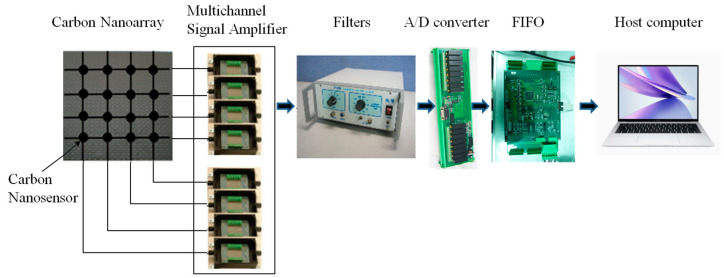
Schematic diagram of damage detection system for 3D braided composite based on carbon nanowires.

**Figure 12 sensors-23-07067-f012:**
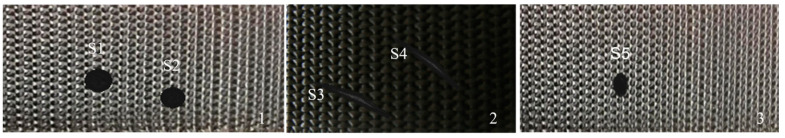
Damage specimens of 3D braided composite embedded in CNT yarns sensors.

**Figure 13 sensors-23-07067-f013:**
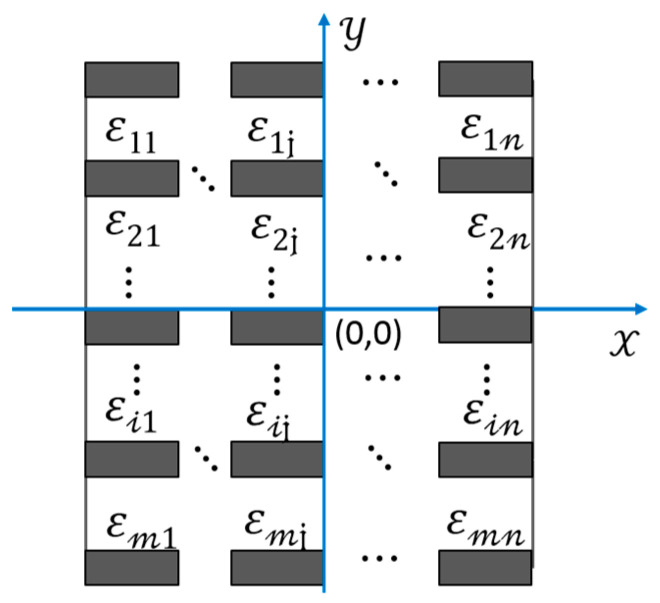
Block diagram of composite material specimen treatment.

**Figure 14 sensors-23-07067-f014:**
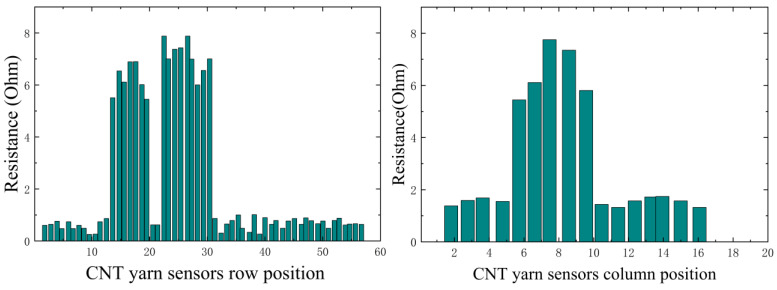
Data detected via CNT yarn sensors in Specimen 1.

**Figure 15 sensors-23-07067-f015:**
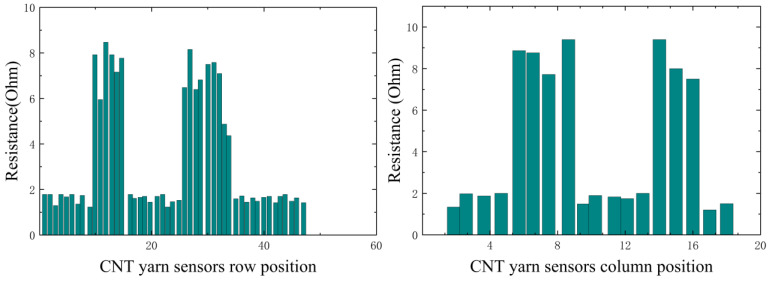
Data detected via CNT yarn sensors in Specimen 2.

**Figure 16 sensors-23-07067-f016:**
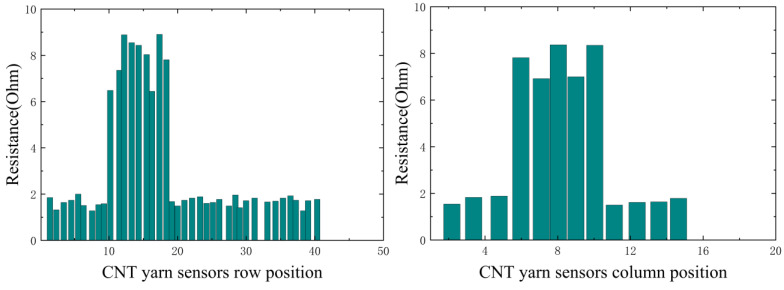
Data detected via CNT yarn sensors in Specimen 3.

**Figure 17 sensors-23-07067-f017:**
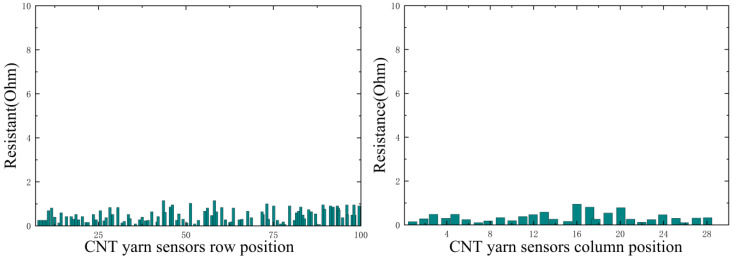
Data detected via CNT yarn sensors in healthy specimens.

**Figure 18 sensors-23-07067-f018:**
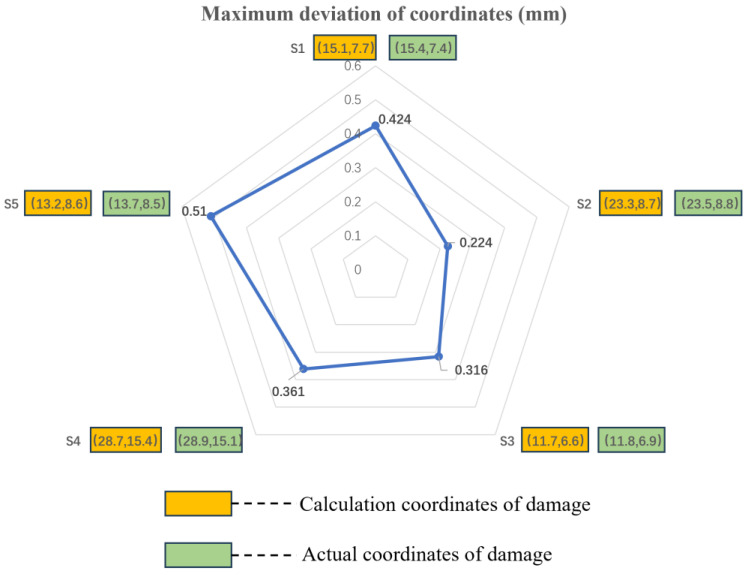
Comparison between detection and actual damage coordinate positions.

**Figure 19 sensors-23-07067-f019:**
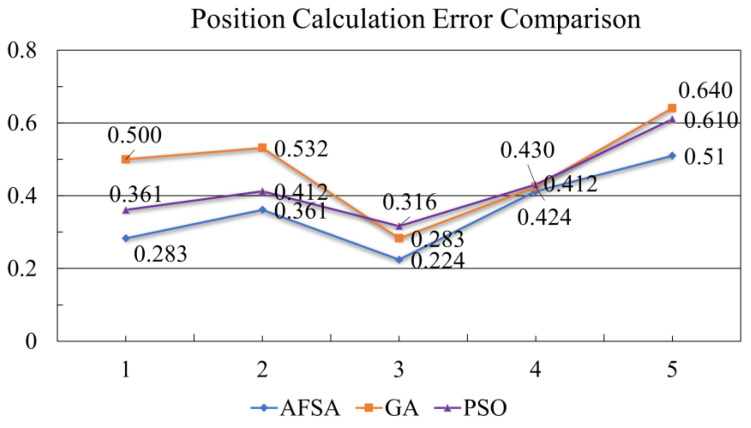
Error comparison diagram of AFSA, GA, and PSO regarding the calculation of damage location.

**Table 1 sensors-23-07067-t001:** Main Technical Parameters of CNT.

Model	Diameter (µm)	Strength (Mpa)	Modulus (Gpa)	Elongation (%)	Density (g/cm^3^)	Electrical Conductivity (S/m)	Length (m)
HQCNTs-014	20–30	270–800	4–6	15–25	0.3–0.5	7 × 10^4^–1 × 10^5^	1–20
HQCNTs-015	5–12	800–1000	50–100	2–3.5	0.3–0.5	5 × 10^4^–7 × 10^4^	1–20
HQCNTs-016	5–12	1000–1200	50–100	2–3.5	0.5–0.8	5 × 10^4^–7 × 10^4^	1–20
HQCNTs-017	5–12	1200–1500	50–100	2.5–3.5	0.8–1.0	5 × 10^4^–7 × 10^4^	1–20

**Table 2 sensors-23-07067-t002:** Configuration of embedded CNT yarn sensor.

Size 180 × 180 × 10 (mm)	Axial sensors	Number	39
		Location	{2, 6,10, 15, 21, 27, 33, 39, 45, 50, 55, 61, 66, 71, 76, 80, 85, 91, 95, 102, 106, 111, 116, 121, 126, 131, 135, 140, 146, 150, 155, 160, 165, 170, 175, 181, 186, 191, 197}
	Weft direction sensors	Number	10
		Location	{2, 6, 8, 11, 14, 17, 21, 25, 29, 34}

## Data Availability

The data presented in this study are available in article.

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
