# Peer review of "Damage Monitoring of Braided Composites Using CNT Yarn Sensor Based on Artificial Fish Swarm Algorithm"

_sensors, 2023, doi:10.3390/s23167067_

Round 1

Reviewer 1 Report

Important study. Some questions to consider.

1. Figs 7, 10 not clear. What do the dots and the lines represent? Is there any interconnection between CNT yarn wires?

2. Line 156, CNT diameter is not tens of microns.

3. Lines 159, 160, an insulating layer is not applied around the CNTs, do you mean the yarn?

4. Give more details of the CNT yarn. Length of CNT and diameter of CNT used to make yarn. Are SWCNT or MWCNT used?

5. Lines 175 to 191. I don't think individual CNT have a poissons ratio like an isotropic material. The CNT are tubes and can have multiple walls and do not behave like an isotropic material. 

6. Line 167. Nanowire, CNT sensor, CNT yarn, and CNT are used somewhat interchangeably. It would be more clear to use one term to define the sensor.

7. Fig 2. Define what is carbon fiber yarn? Is it a tow of carbon fibers? Yarn is usually a twisted fiber material. Won't carbon fiber break if twisted?

8. Line 380 CNT 10 nm cannot be woven itself.

9. Table 1. Using 39+10 CNT yarn sensors can detect damage within 1 mm on a 180 mm square plate seems highly accurate. Are 49 channels of data acquisition needed?

10. Will this technology scale to large structures, eg a wind turbine blade? It seems a very large number of wire connections would be needed.

11. Will multiple layers of CNT sensors be needed for thick structures?

12. Does temperature or humidity affect the sensor accuracy?

13. Fatigue and impact damage could be evaluated to be more realistic.

Nice work.

A few minor formatting line spacing corrections needed.

Reviewer 2 Report

In this manuscript, artificial fish swarm algorithm was utilized to identify damages in braided composites using CNT yarn sensors. Although the manuscript is well written and has interesting issue, some suggestions are provided below to may enrich the scientific level of the manuscript:

-       In the abstract, no information regarding the “braided composites”, the kind of damages, and geometrical details was provided. Moreover, no quantitative result was reported. Please enrich the abstract by adding the required data.

-       It is suggested to add actual photos from the utilized materials and sensors in Section 2.

-       There are a lot of up-to-date publications regarding structural health monitoring and damage detection methods which can be referenced in the introduction section:

·         PdO-Nanoparticle-Embedded Carbon Nanotube Yarns for Wearable Hydrogen Gas Sensing Platforms with Fast and Sensitive Responses. ACS sensors8(1), 94-102.

·         Daneshvar, M. H., Saffarian, M., Jahangir, H., & Sarmadi, H. (2022). Damage identification of structural systems by modal strain energy and an optimization-based iterative regularization method. Engineering with Computers, 1-21.

·         A review of wearable carbon-based sensors for strain detection: Fabrication methods, properties, and mechanisms. Textile Research Journal93(11-12), 2918-2940.

-       All the figures should be replaced with higher quality ones.

-       The font sizes in Figures 8, 9, 13, 14, 15, 16, and 17 can be increased. The frame of Figure 18 can be removed to be consistent to the other figures.

-       It is suggested to add a flowchart at the first part of section 2.3 which shows the process of CNT yarn embedding model based on AFSA. This will help the readers to reuse the proposed method more easily.

-       It is suggested to add more quantitative results to the conclusion section.
